



# Scale-dependent analysis of in situ observations in the mesoscale to submesoscale range around New Caledonia

Guillaume Sérazin[1], Frédéric Marin[2], Lionel Gourdeau[2], Sophie Cravatte[2], Rosemary Morrow[3], and Mei-Ling Dabat[3]

[1]Climate Change Research Centre, University of New South Wales, Sydney NWS 2052, Australia
[2]LEGOS/IRD, Toulouse, France
[3]CNRS/LEGOS, Toulouse, France

**Correspondence:** g.serazin@unsw.edu.au

**Abstract.** Small-scale ocean dynamics around New Caledonia (22°S) in the Southwest Pacific Ocean occur in regions with substantial mesoscale eddies, complex bathymetry, complex intertwined currents, islands, and strong internal tides. Using second order structure functions applied to observational ADCP and TSG dataset, these small-scale dynamics are characterised in the range of scales 3-100 km in order to determine the turbulent regime at work. A Helmholtz decomposition is used to analyse the contribution of rotational and divergent motions. A surface intensified regime is shown to be at work south and east of New Caledonia, involving substantial rotational motions such as submesoscale structures generated by mixed layer instabilities and frontogenesis. This regime is however absent north of New Caledonia, where mesoscale eddies are weaker and surface available potential energy is smaller at small scales. North of New Caledonia and below 200 m in the regions south and east of New Caledonia, the dynamical regime at work could be explained by stratified turbulence as divergent and rotational motions have similar contribution, but weakly nonlinear interaction between inertia gravity waves are also possible as structure functions get close to the empirical spectrum model for inertia gravity waves. Seasonal variations of the available potential energy reservoir, associated with a change in the vertical profile rather than in horizontal density variance, suggest that submesoscale motions would also seasonally vary around New Caledonia. Overall, a loss of geostrophic balance is likely to occur at scales smaller than 10 km, where the contribution of divergent motions become significant.

## 1   Introduction

Oceanic submesoscale (SM) motions lie between horizontal mesoscale (ME) motions O(10-100 km), which are strongly constrained by the Earth's rotation and the ocean's aspect ratio, and three-dimensional turbulent processes occurring at scales smaller than a few hundred meters. The ubiquitous nature of these SM features in the global ocean, such as fronts, filaments and SM coherent vortices, has been demonstrated by high-resolution ocean models, satellite observations and dedicated observational programs (e.g., LatMix, Shcherbina et al., 2014). Although those SM turbulent features represent less kinetic energy (KE) that ME eddies, they may have substantial impacts on heat transport and air-sea heat fluxes (Su et al., 2018), on upper ocean restratification (Boccaletti et al., 2007) and on biogeochemistry processes such as primary production (Mahadevan, 2016; Lévy et al., 2012).



Inertia Gravity Waves (IGWs) are also a key component of the ocean system, sharing similar spatial scales with ME 0(10-100 km) and SM 0(0.1-10 km) motions but at faster timescales. IGWs feedback onto climatic timescales by contributing significantly to shape the diapycnal mixing distribution in the ocean (MacKinnon et al., 2017). IGWs include internal tides that are generated by barotropic tides interacting with steep topographic features, then radiate away from their generation site, and end their lifecycle by breaking and mixing the surrounding water masses. While IGWs have a weak imprint on surface tracer dynamics, both surface velocities and sea surface height (SSH) reflect IGW dynamics as well as balanced motions[1], including ME and SM motions.

Because of this spatial scale overlap between balanced and IGW motions, understanding the small-scale content of high-resolution observations is not trivial. In particular, the next generation of altimeters will use wide-swath radar interferometry in the Ka-band, which will allow the forthcoming Surface Water Ocean Topography (SWOT) satellite to sample sea level spatial scales potentially down to 15 km (Fu and Ubelmann, 2013). Thus, the SWOT satellite will be able to provide more observational information in terms of balanced motions at the short ME range and long SM ranges. Yet, understanding observed features from high-resolution measurements requires us to be able to characterise, and potentially separate, IGWs and balanced motions. SWOT's orbit will also provide richer information on the imprint of internal tides on the SSH and would contribute to improve today's maps of stationary and nonstationary internal tides estimated from altimetry (e.g., Ray and Zaron, 2015; Zaron, 2017; Zhao, 2017, 2018). In preparation for these future spatial observations, global maps of the transition scale between balanced and IGW were achieved by applying an elaborated frequency-wavenumber filter to high-resolution model outputs (Qiu et al., 2018). Tide-resolving submesoscale-permiting ocean models are, however, in their early stages of simulating both balanced and IGW at the same time and need be to confronted with observations.

In addition to observations from satellite and modelling studies, there exist substantial databases of high-resolution in situ observations, which may be useful sources of information on small-scale oceanic features. One possible way to extract and characterise the small-scale content of in situ observations is to use a scale-dependent analysis. In particular, several studies have used spectral analysis through the usual Fourier transform to characterise the small-scale distribution of KE, based on horizontal velocities from Shipboard Acoustic Doppler Current Profiler (SADCP) (e.g., Callies and Ferrari, 2013; Bühler et al., 2014; Rocha et al., 2015; Qiu et al., 2017; Chereskin et al., 2019) and tracer variance from temperature, salinity and chlorophyl measurements (e.g., Hodges and Rudnick, 2006; Cole et al., 2010; Cole and Rudnick, 2012; Kolodziejczyk et al., 2015). This type of characterisation has gained popularity, particularly as the SWOT SSH requirements are defined in terms of spectra. Yet, spectral analysis is limited to data with uniform sampling and often requires *a priori* an interpolation step when applied to in situ observations. In parallel to these spectral analysis, the computation of structure functions (SFs) is a classic method to study the scale dependence of a physical quantity and has been used in early studies of turbulence to confront theories and experimental setups. SFs have been used to characterize the atmospheric KE spectrum in the upper troposphere and lower stratosphere (Cho and Lindborg, 2001) based on opportunity measurements made by commercial aircraft, while providing insights into the direction of KE cascades (Lindborg and Cho, 2001). Until recently, the use of SFs has been less popular in ocean sciences.

---

[1]The term "balanced motions" refers to motions that are in approximate geostrophic or hydrostatic balance





SFs of temperature and salinity have been computed from ARGO floats and provided encouraging physically results (Mc-
Caffrey et al., 2015), especially as the non-uniform sampling of ARGO data renders the use of the Fourier transform impossible.
60 Applying SF analysis to horizontal velocities measured by drifters in the Gulf of Mexico, Balwada et al. (2016) achieved a
characterisation of KE and energy transfers over scales ranging from 10 m to 1000 km; such a spatial range can hardly be
covered using Fourier techniques. They also used the Helmholtz decomposition to separate rotational and divergent motions,
first designed for Fourier analysis by Bühler et al. (2014) and extended to SFs by Lindborg (2015).

Using this Helmholtz decomposition, several studies have separated the contribution of waves and vortices (i.e., balanced
65 motions) in the spectral domain applied on SADCP measurements (e.g., Bühler et al., 2014; Rocha et al., 2015; Qiu et al., 2017;
Chereskin et al., 2019). To do so, the spectrum of potential energy relative to the IGWs needs to be properly resolved by density
observations in the vertical, or assumed using the standard Garret-Munk spectrum (e.g., Munk, 1981). In regions of strong ME
KE, the transition from balanced to unbalanced motions has been well documented using in situ velocity observations, with
transitions occurring at 15 km in the Kuroshio extension (Qiu et al., 2017), 20 km in the Gulf Stream (Bühler et al., 2014),
70 40 km in the Antarctic Circumpolar Current (Rocha et al., 2015). Performing a similar analysis in different regions of the
Northwest Pacific, Qiu et al. (2017) have highlighted the sensitivity of the transition scale to the KE levels, with a longer
transition scale in regions of weak mesoscale activity. For instance, the latter study estimated the transition scale to exceed
200 km in North Equatorial Current. The transition between balanced and unbalanced flows is regularly associated with a
flattening of the spectral slopes, from $k^{-3}$ to $k^{-2}$ in eddy-active regions. However, this flattening does not occur everywhere
in the ocean as the KE spectra computed from ADCP observations in the North Equatorial Current region (Qiu et al., 2017)
and in the Southern California Current (Chereskin et al., 2019) decrease monotonously, approximately following a $k^{-2}$ law.
The KE spectrum of balanced motions may also be depth-dependent leading to a transition scale that varies with depth; how
the KE spectrum varies with depth is regionally dependent (Qiu et al., 2017).

These aforementioned observational studies have focused on regions where internal wave activity is generally weak com-
pared to ME and SM motions. In this study, we focus on the South West Pacific, and more specifically around New Caledonia,
whose geography and regional circulation are described in Figure 1. This area is a peculiar region of the global ocean where
both ME activity and internal tides are substantial. In particular, the top left panel of Figure 2 shows substantial levels of ME
KE, estimated from satellite altimetry maps, south and east of New Caledonia. In these regions, long-lived ME eddies are ubiq-
uitous and travel westward (Keppler et al., 2018). Baroclinic instabilities can extract energy from the vertical shear between the
eastward-flowing surface SubTropical CounterCurrent (STCC) and the underlying westward-flowing South Equatorial Current
(SEC) to feed the ME KE (Qiu and Chen, 2004). The top right panel of Figure 2 shows the stationary part of the M2 internal
tide, also estimated from altimetry (Ray and Zaron, 2015), that reveals tidal hotspots north and south of New Caledonia. The
presence of a complex and steep bathymetry in this region, associated with large barotropic tides, yields substantial internal
tides. Around New Caledonia, the modelling study performed by Qiu et al. (2018) suggests that the transition scale between
balanced and unbalanced motions is everywhere larger than 150 km, on average over the year. This scale may, however, de-
crease below 50 km in some places during late winter and early spring, as balanced motions undergo substantial seasonal
variations south and east of New Caledonia whereas IGWs undergo substantial seasonal variations north of New Caledonia.





Based only on in situ observations, this paper aims at a better understanding of the dynamics involved in this region over the small-scale wavelength range from 3 to 100 km, including both balanced and IGW motions, that are not well captured in
current gridded altimetry products. To do so, we use a scale-dependent analysis based on both SFs, applied to existing in situ datasets including SADCP and TSG measurements. In order to anticipate the future altimetric swath observations and to design joint experiments where in situ observing systems are deployed under SWOT groundtracks, we address here three main issues: (1) How are KE and tracer variance distributed in the range of wavelength scale 3-100 km around New Caledonia? (2) Are there spatial and seasonal variations of those distributions? (3) What kind of dynamical information can we infer from those
distributions?

Section 2 describes the data we used from different observing systems: upper ocean velocities from shipboard ADCPs, surface temperature and salinity from TSGs, and sea level from altimetric data. Section 3 presents the computation of structure functions applied on SADCP and TSG data. Results obtained from SFs computed on horizontal velocities from SADCP are shown in section 4.1. The seasonality of SFs computed on surface tracers from TSG is shown in section 4.2. Section 5 draws the
conclusions and discusses the small-scale characteristics obtained in the 3-100 km range using our scale-dependent approach.

## 2 Data

### 2.1 Horizontal velocities from Shipboard Acoustic Doppler Current Profiler

Horizontal velocity can be reliably measured by Shipboard Acoustic Doppler Current Profilers (SADCP) mounted on the hull of vessels during transits or research cruises. We use the historical transects around New Caledonia gathered into one dataset by
Cravatte et al. (2015) in order to describe the regional circulation. This database consists of a collection of 109 cruises that have acquired measurements of longitudinal $u_{\parallel}$ and transverse $u_{\perp}$ velocities, from 1990 to 2014 (see Table 1 of Cravatte et al., 2015, for a full description). The SADCP data come from different institutes: IRD (73), JASADCP database from the University of Hawaii (11), CSIRO (23), JAMSTEC data center (2). Most of the data are provided by 150 kHz SADCP instruments and cover an average range of 25 to 300 m depth, with an average vertical resolution of 8 meters and an average temporal resolution of 5
minutes. Some SADCP instruments also provide measurements closer to the surface and down to 500 m depth. The available SADCP sections are shown in the bottom left panel of Fig. 2. More details about data quality control and data filtering applied at this stage are given in Cravatte et al. (2015).

The SADCP database has been edited to retain only data meeting certain requirements for computing structure functions. We have only considered high-frequency SADCP: shiptracks with minimal temporal sampling slower than 15 minutes were
discarded from the analysis. The ship cruising speed must be faster than 4 $m.s^{-1}$ (around 8 knots); this condition aims at avoiding the subsampling of internal waves whose average phase velocity is around 1 $m.s^{-1}$ in the ocean. The segments were defined so that the ship travels almost in a straight line, with a $5°$ variation of the ship heading allowed. This condition is not necessary as we will use the assumption of isotropy in the following analyses, but it turned out to be efficient to avoid very long and winding trajectories. A maximum time break of 15 minutes between measurement is allowed within the same segment;
longer time breaks result in the data split into different segments.



## 2.2 Sea surface temperature and salinity from thermosalinograph data

High-resolution near-surface temperature and salinity have been collected by the French Sea Surface Salinity Observation Service (SSS-SNO) from voluntary observing ships (mainly merchant ships) using thermosalinograph (TSG) instruments (Alory et al., 2015). Tracers are measured near surface because sea water is pumped between 5 and 10 meter depth on the side of the hull, depending on the ship configuration. Measurements are made every 15 seconds and are averaged every 5 minutes by taking the median. The average spatial resolution associated with this 5-minute temporal resolution is around 2.5 km for a 20-knot cruise speed. The TSG data can be downloaded from the LEGOS Sea Surface Salinity database[2]. Only delayed-time data flagged as "Good" and "Probably Good" have been extracted and used in this study. Delayed-time data undergo a more severe quality control than real-time data using the TSG-QC software[3]. Flags are automatically attributed and are checked using visual inspection. Corrections of TSG timeseries are applied if needed by comparison with daily water samples and ARGO data. More details about measurements, processing and quality control of TSG data may be found in Alory et al. (2015).

The SSS-SNO network is global but has a better coverage in the tropical Pacific and in the North Atlantic. Thus, this dataset is particularly suited to study high-resolution surface tracers in the South West Pacific. Available sections around New Caledonia are presented in Fig. 2's bottom right panel. Although the temperature and salinity are measured at 5-10 meter depth, we will assume in the rest of the study that they are representative of the tracer behaviour at the top of the mixed layer and we will use the terms sea surface temperature (SST) and sea surface salinity (SSS). We apply the same tests and the same segmentation criteria to the TSG dataset as we do for the ADCP dataset.

## 2.3 Surface to interior oceanic properties from ARGO climatology

The climatologies of temperature and salinity over the first 500 meters of the water column are taken from the Roemmich-Gilson Argo Climatology (Roemmich and Gilson, 2009) computed over the period 2004-2016[4]. This ARGO climatology is used to better characterise and understand the seasonal variations of the surface and interior oceanic properties jointly with the results obtained from the TSG and SADCP datasets. Exponential profiles of the vertical stratification $N(z)$ are fitted to determine an e-folding scale $b$ and a surface extrapolated buoyancy frequency $N_0$. These quantities will be used in the following to determine the empirical model for IGWs.

## 2.4 Regions of interest

Based on large-scale considerations provided by the EKE and M2 climatology maps (Fig. 2's top panels), as well as former studies around New Caledonia, we have selected four regions, delimited by red lines in Fig. 2, having different characteristics in terms of mesoscale activity and internal wave activity.

Region VAUB corresponds to the 100 km wide Vauban Channel between New Caledonia main island and the Loyalty islands (Hénin et al., 1984; Marchesiello et al., 2010; Cravatte et al., 2015). In this channel, a mean southeastward current, extending

---

[2]http://www.legos.obs-mip.fr/observations/sss/
[3]http://www.ird.fr/us191/spip.php?article63
[4]http://sio-argo.ucsd.edu/RG_Climatology.html





to at least 500 m depth, flows against the mean trade winds, overlying a deeper northwestward current. From Fig. 2, this small region has relatively small EKE and weak M2 internal tides. Yet, the interpolated altimetric product might underestimate EKE in this region because of the proximity of coastlines; this will also impact the mapping technique used to produce the M2 climatology. In situ observations have revealed that the Vauban current is strongly modulated at intraseasonal timescales by large ME eddies propagating westward (Cravatte et al., 2015; Keppel et al., 2018). Velocity observations measured by SADCP in this region have shown substantial semi-diurnal currents, likely related to baroclinic tides (Cravatte et al., 2015).

Region ECAL is the large box situated east of New Caledonia, centred on the Southern branch of the SEC. This region exhibits high ME EKE and substantial M2 internal tides. The internal tides in the SCAL region are likely to radiate from surrounding seamounts and islands. Region SCAL is situated south of New Caledonia and comprises high ME EKE as well as strong M2 internal tides, the latter arising partly from the generation site at the southern tip of New Caledonia. As mentioned in the introduction, those regions may involve substantial baroclinic instabilities that partly explains the large levels of EKE (Qiu and Chen, 2004). Long-lived westward travelling ME eddies also passes through those regions and contribute to EKE (Keppler et al., 2018).

Region NCAL is situated north of New Caledonia and west of Vanuatu islands. This region is less energetic than the southern ones in terms of ME activity but has substantial M2 tidal amplitude and is close to several generation sites induced by island topography. There is observational evidence that weak ME eddies may be generated in the lee of the Vanuatu islands (Keppler et al., 2018). The NCAL region excludes the high variability band at 16°S that is suggested to arise from barotropic instability between the North Vanuatu Jet and the surrounding countercurrents (Qiu et al., 2009).

In these regions around New Caledonia, the first mode of the M2 internal wave is dominant and accounts for over 30-50% of the total M2 tidal energy conversion (Vic et al., 2019). The dissipation of M2 internal wave energy is also amongst the strongest in the world's oceans, with a substantial contribution coming from wave-wave interactions (de Lavergne et al., 2019). The amplitude of coherent M2 mode 1 in terms of sea level variations ranges between 1 and 4 cm (see Fig. 2; Ray and Zaron, 2015); the amplitude of mode 2 is much weaker and estimated to be between 1 and 3 mm (Zhao, 2018). In terms of wavelength, M2 mode 1 scales between 120 and 150 km, mode 2 scales between 60 and 70 km and mode 3 scales between 40 and 50 km (Ray and Zaron, 2015; Zhao, 2018)

These regions are well sampled by shipboard ADCP and TSG as shown by Fig. 2's bottom panels. All these four regions will be crossed during the SWOT fast-sampling phase (black lines on Fig. 2's top panels). They are identified as possible case studies to better understand the small-scale SSH signal, jointly with in situ datasets (d'Ovidio et al., 2019).

## 3 Methods

### 3.1 Structure functions

Whilst most studies have used spectral techniques (e.g., Bühler et al., 2014; Rocha et al., 2015; Qiu et al., 2017) to characterize the scale-dependent distribution of horizontal velocities, measured by SADCP along shiptracks, we prefer using structure functions (SFs) as applied to atmospheric velocities measured during aircraft flights (Cho and Lindborg, 2001). The use of SFs is





mainly motivated by the fact that SFs are more suited to study uneven observations in space. Less preprocessing is needed com-
pared to Fourier methods, which often requires interpolation of velocity measurements on equidistant samples, detrending and
windowing. However, SFs may only be applied on physical quantities whose scale-dependent variance decreases gently, that
is for characterising spectra less steep than $k^{-3}$ (Bennett, 1984; Babiano et al., 1985), where $k$ is the horizontal wavenumber.

### 3.1.1 General definition

We define the increment of a given quantity $Q$ (a scalar or a vector field) between a position vector $\boldsymbol{x}$ and a position vector
$\boldsymbol{x} + \boldsymbol{r}$, where $\boldsymbol{r}$ is a separation vector, as:

$$\delta Q(\boldsymbol{x}, \boldsymbol{r}) = Q(\boldsymbol{x} + \boldsymbol{r}) - Q(\boldsymbol{x}). \tag{1}$$

Under the assumptions of homogeneity (i.e., the statistics do not depend on the position $\boldsymbol{x}$) and stationarity (i.e., the statistic
do not depend on time), the second order SF $D_{QQ}$ associated with the variable $Q$ is computed as:

$$D_{QQ}(\boldsymbol{r}) = \langle \delta Q(\boldsymbol{x}, \boldsymbol{r})^2 \rangle_{\boldsymbol{x}}, \tag{2}$$

where $\langle \cdot \rangle_{\boldsymbol{x}}$ denotes the average of $\delta Q$ over all position vectors $\boldsymbol{x}$. We will also use the assumption of isotropy, that is the statistics
do not depend on the orientation of the separation vectors but only on the distance $r = \|\boldsymbol{r}\|$. Therefore, equation (2) will be
directly applied to compute one-dimensional temperature $D_{\theta\theta}(r)$, salinity $D_{SS}(r)$ and buoyancy $D_{\rho\rho}(r)$ SFs by isotropically
averaging over $\boldsymbol{r}$.

Under the previous assumptions, an analytical relationship can be derived between the second order SF of the variable $Q$
and its power spectrum $\Phi(k)$ (Webb, 1964; Babiano et al., 1985):

$$D_{QQ}(r) = 2 \int_0^\infty \Phi(k) \left[1 - J_0(kr)\right] dk, \tag{3}$$

where $J_0$ is the zeroth order Bessel function of the first kind. If one assumes that the power spectrum $\Phi(k)$ decreases propor-
tional to $k^{-n}$ with $1 < n < 3$, then equation (3) may used to derive that the associated SF $D_{QQ}(r)$ increases proportional to
$r^{n-1}$ (Webb, 1964; Babiano et al., 1985; McCaffrey et al., 2015). In case of strong nonlocal dynamics, the power spectrum
is steeper than $k^{-3}$ and SF saturates at $r^2$, making it impossible for SFs to distinguish between different regimes involving
nonlocal dynamics.

### 3.1.2 Total, transverse and longitudinal velocity SFs

In order to analyse the scale-dependent distribution of horizontal KE, the computation of the total second order velocity SF
(VSF) $D_{\boldsymbol{UU}}$ will be deduced from the longitudinal $D_{\|\|}$ and transverse $D_{\perp\perp}$ second order VSF, isotropically averaged over $\boldsymbol{r}$,
so that:

$$D_{\boldsymbol{UU}}(\boldsymbol{r}) = D_{\|\|}(\boldsymbol{r}) + D_{\perp\perp}(\boldsymbol{r}). \tag{4}$$




These longitudinal and transverse VSFs are computed using a similar definition as in Babiano et al. (1985):

$$D_{\|\|}(\boldsymbol{r}) = \frac{\langle \|\delta \boldsymbol{U}(\boldsymbol{x},\boldsymbol{r}) \cdot \boldsymbol{r}\|^2 \rangle_{\boldsymbol{x}}}{\|\boldsymbol{r}\|^2}, \tag{5}$$

$$D_{\perp\perp}(\boldsymbol{r}) = \frac{\langle \|\delta \boldsymbol{U}(\boldsymbol{x},\boldsymbol{r}) \times \boldsymbol{r}\|^2 \rangle_{\boldsymbol{x}}}{\|\boldsymbol{r}\|^2}. \tag{6}$$

This definition is particularly useful because it does not depend on the coordinate system used and therefore does not require to rotate velocity components prior to the calculation of each increment $\delta \boldsymbol{U}$. The longitudinal and transverse VSFs will be computed on zonal and meridional velocities from the SADCP dataset presented in section 2.

### 3.1.3 Helmholtz decomposition

The Helmholtz decomposition has fundamental applications in fluid dynamics as it provides a mathematical way to decompose
the velocity field as the sum of two components, one purely rotational and one purely divergent. This Helmholtz decomposition has recently been extended to be used in the spectral domain (Bühler et al., 2014) and with VSFs (Lindborg, 2015). Balanced motions - in geostrophic or hydrostatic approximate balance - are mainly associated with rotational motions. IGWs or stratified turbulence have, however, a strong divergent component and may also have a non-negligible rotational component.

Under the assumption of homogeneity and isotropy, Lindborg (2015) showed that this separation between rotational and
divergent motions is straightforward using VSFs. The rotational VSF $D_{\psi\psi}$ and the divergent VSF $D_{\phi\phi}$ are simply derived from two integrals of the longitudinal $D_{\|\|}$ and transverse $D_{\perp\perp}$ VSFs:

$$D_{\psi\psi}(r) = D_{\perp\perp}(r) + \int_0^r \left[ D_{\perp\perp}(r') - D_{\|\|}(r') \right] dr', \tag{7}$$

$$D_{\phi\phi}(r) = D_{\|\|}(r) - \int_0^r \left[ D_{\perp\perp}(r') - D_{\|\|}(r') \right] dr'. \tag{8}$$

Thus, we will use a total of three quantities deduced from the measured horizontal velocities to interpret the scale-dependent
distribution of KE: total, rotational and divergent VSFs. The spectral slopes associated with the total VSF will also be computed as well as the mean ratio of the divergent and rotational VSFs $R_{\phi\psi} = D_{\phi\phi}/D_{\psi\psi}$.

### 3.2 Classic power laws and relevant turbulence theories

In addition to the Helmholtz decomposition, power laws estimated on SFs and spectra can be used to infer the dynamics at work. However, there are two limitations in predicting the turbulence regime using only the slopes of the KE spectra. On one
hand, a typical value of a spectral slopes may correspond to different dynamical regimes. On the other hand, several dynamical regimes can combine or overlap at a certain range of scales, in our case, balanced motions and IGWs. In this section, we briefly review the different theories available to describe turbulent regimes occurring in the ocean by focusing on the corresponding spectra/SF scaling laws for KE and tracer variance as well as the ratio of divergent and rotational VSFs $R_{\phi\psi}$.





### 3.2.1 Kinetic energy

The first scaling law for KE was introduced by Kolmogorov (1941) for three-dimensional isotropic homogeneous turbulence. Using a dimensional analysis, he predicted an inertial range where KE decreases proportional to $k^{-5/3}$ ($r^{2/3}$) corresponding to a forward cascade of KE, eventually leading to dissipation at small scales. A similar analysis was later performed for two-dimensional horizontal turbulence (Kraichnan, 1967) yielding two inertial ranges: a $k^{-5/3}$ ($r^{2/3}$) range corresponding to an inverse cascade of KE and a steeper $k^{-3}$ ($r^2$) range corresponding to a forward cascade of enstrophy. When simple stratification

is added to the quasi-geostrophic equations, baroclinic instability becomes the main source of KE around scales close to the internal deformation radius, yet turbulence behaves as in two-dimensional flows with similar inertial ranges (Charney, 1971; Salmon, 1998). For quasi-geostrophic dynamics, the ratio $R_{\phi\psi}$ is expected to be small because rotational motions dominate.

Quasi-geostrophic turbulence, however, should hold only for balanced motions that are characterised by a strong rotational component and a Rossby number $Ro \ll 1$. At smaller scales, geostrophic balance weakens and the contribution of ageostrophic

components, having both rotational and divergent components, becomes substantial. In particular, frontogenesis processes are accelerated by ageostrophic motions (Hoskins and Bretherton, 1972) and are predicted to yield $k^{-2}$ ($r^1$) spectra (Boyd, 1992). At similar scales, i.e. 1-10 km, symmetric and baroclinic instabilities occurring in the mixed layer (e.g., Boccaletti et al., 2007) substantially energise submesoscale motions near the surface and also yield KE spectra close to $k^{-2}$ (Callies and Ferrari, 2013).

Overlapping with the previous regimes, IGWs span frequencies from inertial $f$ to buoyancy $N$ and are associated with a

wide range of horizontal scales, tied to the previous frequencies by the IGW dispersion relation. Since the seminal work of Garrett and Munk (1972), the IGW continuum spectrum has been shown to be a robust feature in the global ocean and can be estimated by the Garrett and Munk (GM) empirical model (e.g., Munk, 1981). This model shows that IGW spectrum scales as $k^{-2}$ in the short-wave limit but flattens out at scales larger than 10 km (see also Callies and Ferrari, 2013). Since our scales of interest is in the range 1-100 km, we cannot infer IGWs directly from spectral slopes and we use instead the GM model

spectrum converted into structure functions using (3). One may also distinguish waves with frequencies $\omega \gg f$ that yield a ratio $R_{\phi\psi} \gg 1$ and near-inertial waves with frequencies close to $f$ that have $R_{\phi\psi} \gtrsim 1$ (Li and Lindborg, 2018) .

Stratified turbulence is also a turbulence regime possibly at work at small scales, that yields KE spectra close to $k^{-5/3}$ ($r^{2/3}$), associated with a forward cascade of KE involving contributions from ageostrophic components (Lindborg, 2006), and characterised by $R_{\phi\psi} \sim 1$ (Lindborg and Brethouwer, 2007; Li and Lindborg, 2018). Stratified turbulence has been suggested

by Lindborg (2015) to explain why the observed atmospheric SFs scale as $r^{2/3}$ at small scales.

### 3.2.2 Tracers and potential energy

Following Kolomogorov-like arguments, Obukhov (1949) and Corrsin (1951) predicted that the spectrum of tracer variance decreases with the similar rate of $k^{-5/3}$ ($r^{2/3}$) as KE does in the direct cascade range for quasi-geostrophic turbulence. More generally, Vallis (2006) reviews how KE and tracer variance spectra are linked for KE spectral slopes less steep than $k^{-3}$ ($r^2$).

For steeper KE spectra, the stirring is nonlocal and dominated by a single eddy-turnover time scale and yields a tracer variance spectrum rolling off as $k^{-1}$ ($r^0$), characteristic of a Batchelor spectrum (Batchelor, 1959; Vallis, 2006). On the contrary surface





frontogenesis, stirring active and passive tracers locally, yields a spectral distribution tracer variance following a logarithmic slope of $k^{-2}$ (Klein et al., 1998), i.e., $r^1$ for tracer SFs. Other surface intensified processes involving substantial ageostrophic flows such as mixed-layer dynamics may also explain this $k^{-2}$ ($r^1$) power laws (Callies and Ferrari, 2013). Finally, the internal-
wave continuum induces similar spectral/SF slopes (Callies and Ferrari, 2013), for scales smaller than 10 km, but slopes flattens out at larger scale. The imprint of internal waves on surface tracers is, however, expected to be weak.

## 4  Results

We present here the results of the SFs applied on velocity and surface tracer data around New Caledonia. First, the VSFs are used to discuss the dynamical regime at work in the upper 500 meters of the ocean. As not enough ADCP data are available to
capture the annual cycle with the VSFs, the tracer SFs are then used to discuss of the seasonality of upper ocean dynamics.

### 4.1  Velocity structure functions

Second order SFs are computed on longitudinal and transverse velocity components for the four regions of study on each ADCP shiptrack. The VSFs are averaged over different depth ranges: a surface layer 0-100 m, a transition layer ranging from 100 m to 200 m, an ocean interior layer ranging from 200 m to 500 m. Figures 4 and 5 show the SFs over these three layers
for the four regions of study for the range of spatial scales 3-100 km. The Helmholtz decomposition is eventually performed on every layer to separate the contribution of the rotational and divergent components. These operations are performed on each ADCP segments, then the median is computed. Confidence intervals at 5% and 95% are estimated using a bootstrap method by performing 10000 realisations of randomly chosen ADCP segments.

In the VAUB region, the surface layer shows that the total VSF (green curve) has a slope close to 1 (1.01) consistent with
surface intensified dynamics (Fig. 4's top left panel). The total VSF slope tends to decrease with depth to get closer to the power law $r^{2/3}$ in the ocean interior (slopes of 0.76 and 0.56 respectively for the 100-200 m and 200-500 m ranges shown in middle and bottom panels of Fig. 4). In the surface layer, the rotational VSF (red curve) contributes significantly to most of the total VSF over the 3-100 km range and clearly dominates the divergent SF (purple curve). Deeper in the water column (middle and bottom panels of Fig. 4), the rotational VSF still dominates in the 10-100 km range but its contribution to the total VSF drops
in the 3-10 km range as the divergent SF contribution increases. The contribution of rotational and divergent motions become equivalent under 10 km in the 100-200 m layer and around 10 km in the 200-500 m layer, albeit with large uncertainties for the latter layer due to the quality of the data. In the 200-500 m layer, the shape of the total VSF more closely follow the shape of GM SF (black curve in bottom left panel), while the average ratio $R_{\phi\psi}$ is 0.6 (compared to 0.3 for the other layers).

Thus, VSF analysis shows that the VAUB region is consistent with a surface-intensified regime, dominated by rotational
dynamics over the ME to SM range, with a VSF slope consistent with frontogenesis and mixed layer dynamics. These surface dynamics contrast with the ocean interior as divergent dynamics become more substantial with depth and the slope of the total VSF tend to flatten towards $r^{2/3}$. Although the VSFs get close to the shape of the GM SF, the divergent VSF never exceeds the rotational VSF in the interior as the ratio $R\phi\psi$ is smaller than unity. Stratified turbulence is then more likely to be at work





than weak interactions between IGWs to yield such VSF shape. The interior layer (bottom left panel) is, however, uncertain

because less valid data is available at those depths as shown by the large confidence intervals in Fig. 4's bottom left panel.

The VSF functions computed on SADCP data in the ECAL region (Fig. 4's right panel) exhibit similar results than the VAUB region. The slope of the total VSF decreases with depth from 0.87 at surface (0-100 m) to 0.65 in the interior (200-500 m). The rotational motions (red curve) also dominates at surface but their relative contribution to the total VSF decreases with depth as divergent motions (purple curve) become more substantial. In the transition layer (middle left panel), divergent and rotational

VSF has a similar amplitude in the range 3-10 km whereas rotational motions still dominates the range 10-100 km, yielding an average ratio $R_{\phi\psi}$ of 0.49. VSFs computed in the interior layer (bottom left panel) are less precise, with large confidence intervals, but they suggest that divergent motions have a substantial impact over the whole range of spatial scales (3-100 km) as $R_{\phi\psi}$ gets slightly larger than unity (1.55). In this interior layer, the total VSF also ressembles to the GM structure function suggesting, jointly with the previous value $R_{\phi\psi}$, that weak interactions between IGWs are possibly at work to generate the

observed VSFs. As $R_{\phi\psi}$ is close to unity, nonlinear interactions due to stratified turbulence could also be at work.

The NCAL region (Fig. 5's left panel) distinguishes from the ECAL and VAUB regions as the slope of the total VSF is already close to the $r^{2/3}$ law at the surface (0.71) and consequently decreases with depth to a lesser extent (to a value of 0.49). At the surface, rotational and divergent VSFs have a similar magnitude over the range 3-100 km with $R_{\phi\psi}$ being equal to 1.07. Rotational and divergent motions have also a similar contribution in the intermediate layer ($R_{\phi\psi}$=0.95), albeit with weaker

variance that also impacts on the slope of the total VSFs (green curve). Averaged results in the ocean interior (200-500 m) are not qualitatively different but SFs (grey curves) are noisier due to data of lesser quality at these depths. Unlike the ECAL and VAUB regions, surface dynamics in the NCAL region seem to be weak or potentially masked by more substantial unbalanced dynamics. The values of $R_{\phi\psi}$ close to unity suggest that stratified turbulence might be at work but weak interactions between IGWs with frequencies close to $f$ might also be responsible for the observed VSFs. The ressemblance between the shape of

the GM VSF and the total VSF, especially in the interior layers, also argues that observed VSFs could be consistent with IGW interactions.

In the SCAL region (Fig. 5's right panel), the surface and intermediate layers show a total VSF with slopes close to 0.8, between the 1 and 2/3 power laws. Rotational motions (red curve) dominate over the whole range of scales (3-100 km), involving a substantial role of SM and ME processes, albeit the VSF slope is not fully consistent with frontogenesis and

mixed layer dynamics. In the ocean interior (200-500 m, bottom panel), the slope of the total VSF slightly decreases to 0.76 as divergent motions becomes substantial in the 10-100 km range and equivalent to rotational motions in the 3-10 km range, yielding an average $R_{\phi\psi}$ of 0.8 (compared to 0.4 for the other layers). In this layer, the regime at work is not clear because rotational motions has a slightly larger contribution to the total VSF than divergent motions. The equivalent contribution of rotational and divergent motions at smaller scales suggests, however, that stratified turbulence could be at work. In this region,

the contribution of divergent motions to the total VSF might keep increasing with depth as seen in the other regions.

Figs. 4 and 5 show that the total VSFs tend to flatten with depth in each of the four regions, albeit this decrease is less clear for the NCAL and SCAL regions. To illustrate this tendency, we compute averaged slopes for each total VSFs over 30 meter bins of depth. Slopes less than 0.2 were discarded from the analysis because they are likely to be due to instrumental





noise (i.e., a slope close to 0 characterises an uncorrelated signal). Results are plotted in Figure 6's left panel and clearly show
a dependence with depth for the four regions. This depth dependence also supports our decomposition into a surface layer
(0-100 m) where the VSF slope is relatively constant between 0.75 and 1, a transition layer where the VSF slope rapidly drops
(100-200 m), and the interior layer where the VSF slope slightly decreases to reach values between 0.5 and 0.6. Note that the
uncertainty on the VSF slopes at depth are stronger because less valid data is available. As previously noticed, Fig. 6's left
panel also shows that the transition with depth is less marked for the NCAL region.

In order to assess the scales at which the quasi-geostrophic balance is likely to hold, we infer the scale-dependent Rossby
number from the total VSF as

$$Ro(r) = \frac{\sqrt{D_{UU}}}{fr}. \tag{9}$$

The scale-dependent Rossby number is computed in each region and averaged over all depth. Fig. 6's right panel shows that
$Ro$ logically increases as the scale $r$ decreases, with $Ro$ being close to unity at the kilometric scale for all regions of study. At
scales larger than 10 km, the Rossby number is large enough ($> 10$) to consider that rotation effects dominate the dynamics
and suggests that total VSFs with large rotational components are associated with quasi-geostrophic eddies at those scales.
Below 10 km, a loss of geostrophic balance is likely to happen and substantial ageostrophic motions would imprint on velocity
divergence. This assessment is confirmed by a substantial contribution of the divergent VSFs (Figs. 4 and 5) at scales smaller
than 10 km for all the layers and regions.

In summary, the analysis of VSFs computed on SADCP measurements suggests that there is a surface intensified regime at
work in the VAUB and ECAL regions, and to a lesser extent in the SCAL region. Close to the surface (0-100 m), such a regime
is in accordance with mixed layer dynamics and frontogenesis in the VAUB and ECAL regions, involving quasi-geostrophic
vortices at scales > 10 km with a predominant rotational component whereas smaller scales are likely to undergo a loss of
balance as the contribution of divergent motions increases. In the interior layer (200-500 m), divergent motions have generally
equivalent amplitude as rotational motions, suggesting that stratified turbulence could be at work in the ocean interior. The
hypothesis of a regime involving weak interactions between IGWs is also plausible as the shape of the total VSFs resembles
that of the GM VSFs in some regions. Finally, the NCAL region does not show a surface intensified regime, except for a slight
transition of the slope of the total VSFs. The NCAL region seems to be rather impacted by IGWs in all the layers studied, in a
similar way as the interior layer of the other regions.

The presence of substantial rotational motions in the ECAL and SCAL regions are consistent with substantial ME EKE
estimated from altimetry and associated with baroclinic instabilities and westward travelling ME eddies in those regions (see
section 2). There is a discrepancy between the EKE map in the VAUB region and the importance of rotational motions in
this region, perhaps due to an underestimate of sea level variance by mapping altimetric data close to the coast. The relative
importance of divergent motions compared to rotational motions in region NCAL is also consistent with weak ME EKE and
substantial coherent internal tides in this region.





## 4.2 Tracer structure functions

To provide complementary insights into surface small-scale dynamics and analyse their seasonal variations, second order SFs are now computed on each segment of surface tracers measured by shipboard TSGs in the regions NCAL, ECAL and SCAL. The TSG coverage over the VAUB region does not allow an accurate computation of surface tracer SFs, which are therefore

not shown. Figure 7 shows the SFs computed on surface temperature ($SST$, blue curves panel), salinity ($SSS$, red curves), and density ($SS\rho$, green curves) from 3 km up to 1000 km for each regions. The SFs of $SST$, $SSS$, $SS\rho$ have been respectively adimensionalised by $\alpha^2$ (thermal expansion coefficient), $\beta^2$ (haline contraction coefficient), and $1/\rho_0$, $\rho_0 = 1025\,kg.m^{-3}$ being the reference density of sea water. The coefficients $\alpha$ and $\beta$ were computed between consecutive TSG observations and averaged over each segment. This normalisation allows the comparison of the different tracer SFs on the same plot. The SFs

of each segment are averaged over two different seasons of the year: one cool season (solid curves) from June to November and one warm season (dashed curves) from December to May. As for VSFs, confidence intervals at 5% and 95% are estimated using a similar bootstrap method.

First, the dynamical regimes can be discussed in light of the tracer SF slopes, estimated over the range 3-100 km and given in the legend of Fig. 7. The slopes of the temperature SFs are close to 1.2 for the two seasons in all the regions, except

during the warm season in the NCAL region where it reaches 1.33. The temperature SFs are slightly steeper than those of salinity SFs, whose slope are comprised between 0.93 to 1.07, close to the $r^1$ law. The slopes of density SFs mostly reflect the slopes of temperature SFs and range from 1.09 to 1.21. Surface tracer SFs are therefore consistent with stirring induced by frontogenesis processes and submesoscale motions in the ECAL and NCAL regions, albeit the SST and density SFs are slightly steeper than theoretical prediction (i.e., $r^1$ slope). Those tracer SF slopes are also consistent with those of the VSFs

presented earlier principally in the ECAL region. The discrepancies between tracer and velocity SFs in the NCAL and the SCAL regions could be explained by the substantial IGWs occurring in these regions that possibly have a significant imprint on horizontal velocities near the surface but not on surface tracers. No substantial seasonal variation is noticed on the slope of the tracer VSFs, meaning that the rate of stirring by mesoscale and submesoscale structures is not seasonally dependent. Note that tracer SFs in the SCAL region are more uncertain as shown by the large confidence intervals in Figure 7's bottom panel

due to less TSG segments available.

The quality and the quantity of the TSG measurements allows the computation of tracer SFs up to scales as large as 1000 km, providing additional information in the range 100-1000 km that was missing in the VSF analysis. For scales larger than 100 km, the tracer SF slopes seem to follow the same power law for the NCAL and SCAL regions (Fig. 7's top and bottom panels) whereas the slopes tend to flatten in the ECAL regions (middle panel) towards a zero slope. A $r^0$ slope may indicate that the SF

is uncorrelated at larger scales, perhaps because the assumption that turbulent statistics can be sampled with a moving vessel fails at those scales. However, a $r^0$ slope is also compatible with nonlocal stirring of passive tracers by large-scale dominant eddies (Charney, 1971; Callies and Ferrari, 2013). The latter hypothesis is consistent with a similar flattening of tracer spectra (from $k^{-2}$ to $k^{-1}$) shown by Kolodziejczyk et al. (2015) in the subtropical North Atlantic using TSG data. Note that the ECAL





region is particularly impacted by westward-propagating mesoscale structures, which are likely to follow quasi-geostrophic
behaviour and explain large-scale stirring with $r^0$ ($k^{-1}$).

Although there is no significant seasonal variation of the tracer SF slopes, a striking feature of Figure 7 is the seasonal
variations of the amplitude of salinity and temperature SFs that are significative in the NCAL and ECAL regions, albeit with
an opposite behaviour. Temperature SFs have larger amplitude during the cool season as opposed to the warm season, whereas
salinity SFs have weaker amplitude during the cool season compared to the warm season. The seasonal variations of the
density SFs are much weaker and cannot be properly disentangled because of the associated uncertainties; the median however
suggests a similar seasonality as the temperature SFs. Because neither the temperature SFs nor the salinity SFs exceed the
amplitude of the density SFs, no compensation between salinity and temperature is at work in those regions. On the contrary,
there are constructive effects between salinity and temperature, whose intensity depends on the season; stronger constructive
effects between temperature and salinity occur during the warm season. The tracer SFs in the SCAL region are noisier, with
large confidence intervals that hamper the detection of a significant seasonality. The median tracer SFs however suggest similar
seasonality as noticed in the NCAL and ECAL regions.

To explore further the relative importance of temperature and salinity over spatial scales, the scale-dependent absolute density
ratio $|R(r)|$ may be inferred directly from the temperature and salinity SFs:

$$|R_\rho(r)| = \frac{\alpha}{\beta}\sqrt{\frac{D_{\theta\theta}(r)}{D_{SS}(r)}} \tag{10}$$

This ratio is shown on Figure 8's left panel for the three regions of study averaged during the cool and warm seasons. For
all those cases, the density ratio $|R(r)|$ is larger than unity over all scales, meaning that temperature effects dominate over
salinity effects, except for the NCAL region during the cool season where both effects are equivalent ($|R| \sim 1$). The density
ratio also tends to decrease with smaller scales, meaning that salinity effects on density become more important at small scales.
At scales larger than 200 km, discrepancies between regions start to appear but less data are available at those scales, hence
more uncertainties. Figure 8's left panel also exhibits a seasonality of the density ratio with larger $|R|$ values during the cool
season and smaller $|R|$ values during the warm season. Thus, the relative contribution of temperature and salinity to density
varies with seasons.

Can we link the seasonal variations of tracer SFs and those of the vertical hydrographic profiles? In the NCAL and ECAL
regions, Figure 3 shows that the upper 100 meters of the ocean also undergo seasonal variations. During the cool season,
salinity close to the upper 50 meters tends to increase in relation with the seasonality of the South Pacific Convergence Zone
(e.g., Delcroix, 1998), leading to a reduction of the salinity stratification in the upper 100 meters. This surface saltening during
the cool season is associated with weaker salinity variance at small scales as shown by the smaller amplitude of salinity SFs (red
curves in Figure 7). In the SCAL region, the weak seasonal variation in the salinity profile is associated with weak seasonality in
salinity SFs during this season. The opposite effect happens for upper ocean temperature. While temperature and its associated
stratification are reduced in the upper 100 meters during the cool season, in relation with the seasonal variations of the solar
input and the mixed layer, small-scale temperature variance increases as shown by the larger amplitude of temperature SFs
during this season (blue curves in Figure 7). Although the upper ocean density tends to decrease during the warm season as





stratification increases (bottom panels), the density SFs do not seem to be influenced by this seasonal variation, contrary to temperature and salinity. Thus, no clear dynamical link between seasonal variations of the hydrographic profiles and tracer SFs

seems to emerge from this analysis.

Although the seasonality of EKE cannot be inferred from the VSFs because of the lack of data, the seasonality of the EKE counterpart, that is the Available Potential Energy (APE), can be inferred from the density SFs and the vertical profile of $N$. The corresponding SF for the APE near the surface is estimated as:

$$D_{APE}(r) = \frac{g^2}{N^2} \frac{D_{\rho\rho}(r)}{\rho_0^2}, \tag{11}$$

where $g$ is the gravitational acceleration. As TSG measures temperature and salinity between 5 and 10 meter depth, $N$ is taken averaged in this depth range from the ARGO climatological mean profiles for each region. The APE represents the reservoir of energy that can be converted into EKE by oceanic instabilities. The APE SFs are shown in Fig. 8's right panel for the three regions and for the two seasons. First of all, there are substantial seasonal variations with more APE during the cool season (solid curves) than during the warm season (dashed curves). The finding of seasonal variation of the APE in the ECAL

and SCAL regions is consistent with the seasonal modulation of the EKE in the South Pacific Subtropical Countercurrent (STCC) shown by Qiu and Chen (2004) using altimetric data: EKE increases during the cool season (June-November) to peak in November/December whereas EKE decreases during the warm season (December-May) to its minimum in June. The same authors also show that the cool season is more favorable to baroclinic instability, which is a necessary condition to release energy from a larger APE reservoir during the cool season. Since we have seen earlier that the seasonality of the surface

density SFs is weak, most of this APE seasonality is due to seasonal changes in the stratification (see Fig. 3). The APE is also smaller in the NCAL region compared to the ECAL and SCAL regions, consistent with larger EKE in the latter regions as shown in Fig. 2's top right panel. The seasonal and spatial variations noticed on large mesoscale EKE captured by altimeters may also be valid for smaller scales as the APE follow similar seasonal and spatial variations in the range 3-100 km.

In summary, the SFs applied on surface tracers are in accordance with submesoscale dynamics (e.g., mixed layer dynamics,

frontogenesis) as slopes are close to $r^1$ over the range 3-100 km. In the ECAL region, the slopes of tracer SFs are also consistent with those of VSFs for the surface layer. Differences are noticed for scales larger than 100 km that are either due to a lack of quality data (NCAL and SCAL) or to a different dynamical regime (ECAL). Temperature and salinity SFs exhibits seasonal variations in phase opposition, and are associated with seasonal modulation of constructive effects yielding to a weak seasonality in the density SFs. No clear dynamical link is established between the seasonal variations of temperature and

salinity SFs and those of hydrographic profiles. Contrary to the density SFs, the APE SFs show seasonal variation mainly due to the seasonality of the stratification that could yield seasonal variation in surface EKE and VSFs. Spatial and seasonal variations of APE in the range 3-100 km are consistent with larger scale variation of EKE observed by altimetry.



## 5   Conclusion and discussion

In this study, we characterised the distribution of Kinetic Energy (KE) and tracer variance among the range of scales 3-100
km around New Caledonia using structure functions (SFs). We gathered data from different observing systems around New
Caledonia, including SADCP, TSG, ARGO, which made this study purely based on in situ observations. The analyses of SFs
computed on these data showed that the dynamical regime at work in the 3-100 km range depends on depths, regions, seasons
and scales. For each dependence type, we dedicate a section in which our findings are summarised and discussed. We eventually
give some perspectives arising from this study.

### 5.1   Depth dependence

First, we found that different dynamical regimes are likely to be at work between the surface and the ocean interior in the
regions south (SCAL) and east (ECAL) of New Caledonia as well as in the Vauban (VAUB) channel. Our results in those
regions suggest the existence of a surface intensified regime consistent with submesoscale dynamics involving frontogenesis
and mixed layer instabilities. This hypothesis is supported by total velocity and tracer SFs having slopes close to $r^1$ ($k^{-2}$) near
the surface, with motions involving a dominant rotational component. However, the importance of rotational motions weakens
with depth at the same time as the total VSF slopes flatten out to get closer to the $r^{2/3}$ ($k^{-5/3}$) law. Because the average ratio
of divergent and rotational components $R_{\phi\psi}$ is close to unity at these depths, we concluded that stratified turbulence could be
a regime at work in the interior layer (200-500 m). However, the shape of the total VSF at these depths is also close to the GM
SF, which suggests together with $R_{\phi\psi}$ that the turbulent regime could consist in weak nonlinear interactions between IGWs.
As noticed by Qiu et al. (2017) in some regions of the North Pacific, we found that the turbulent regime is depth dependent
with rotational motions decreasing in the ocean interior. What controls the depth-dependence of the turbulent regime we found
is unclear, but there are reasons to think that the depth of the mean currents might have an impact. Our study has not addressed
the processes at work in the generation of submesoscale structures that could explain this surface intensified regime, including
mixed layer instabilities, strain-induced frontogenesis and the turbulent thermal wind balance (McWilliams, 2016; Srinivasan
et al., 2017). Those instabilities extract their energy from the mesoscale gradients of buoyancy, which are substantial south and
east of New Caledonia. In addition to observational datasets, a modelling approach to simulate submesoscale dynamics and
IGWs around New Caledonia would be valuable to characterise in details the submesoscale processes at work and investigate
this depth dependence.

### 5.2   Regional dependence

Secondly, our analysis exhibited regional discrepancies between the SCAL and ECAL regions, which include the surface
intensified STCC current flowing eastward associated with substantial levels of EKE, and the NCAL region, which has weaker
EKE but substantial internal tides. In particular, we did not find a clear surface intensified regime in the NCAL region. Instead,
we found that IGWs might already dominate near the surface with similar contributions of rotational and divergent motions
across the submesoscale-to-mesoscale range as well as a slope close to $r^{2/3}$ ($k^{-5/3}$). We also found that the NCAL region has





lower APE in this range of scales, consistent with a weaker EKE and weaker submesoscale motions associated with a large rotational component. In the interior layer (200-500 m), we also found that rotational motions still predominate at scales larger than 10 km in the VAUB region contrary to the others. At scales larger than 100 km, the tracer SFs in the ECAL region flatten out and departs from the other regions, suggesting a regime closer to interior quasi-geostrophic turbulence, whose stirring is non locally generated by large-scale eddies.

The location of hotspots of ME EKE and coherent internal tides shown in Fig. 2 seems to be linked with the dynamical regimes occurring in the range 3-100 km and supports the decomposition of the SF analyses into different regions. This regional link partly exists because coherent internal tides are associated with substantial non-coherent smaller IGWs and ME EKE provides APE for submesoscale structures. In some regions around New Caledonia, we also expect that some submesoscale structures can be generated in the lee of islands due to horizontal shear instabilities of the mean currents (McWilliams, 2016; 515 Srinivasan et al., 2019): Keppler et al. (2018) found a substantial amount of eddies in the wake of the Vanuatu islands and north of New Caledonia where the lagoon ends using an eddy-tracking algorithm on altimetric sea level maps.

Because of those regional peculiarities, the turbulent regimes around New Caledonia characterised by SFs within the range 3-100 km are substantially different from those occurring at midlatitude eddy-active regions (e.g., Bühler et al., 2014; Rocha et al., 2015; Qiu et al., 2017). We did not find spectra transitioning from $k^{-3}$ to $k^{-2}$ near surface, characterising a transition 520 between balanced and unbalanced flows. Instead, we found slopes close to $k^{-2}$ in three regions (ECAL, SCAL, VAUB), as in regions of weaker EKE such the North Equatorial Current region (Qiu et al., 2017) and the southern California Current System (Chereskin et al., 2019).

With the region north of New Caledonia (NCAL), we also provide the first observational evidence that the KE spectrum may already be close to $k^{-5/3}$ ($r^{-2/3}$ for VSFs) in the range 3-100 km, with a substantial contribution of divergent motions. In the 525 Gulf of Mexico, VSFs were shown to have an inertial range of $r^{-2/3}$ but at scales smaller than 1 km, and a slope close to $r^1$ in the range 1-100 km (Balwada et al., 2016). Thus, the NCAL region is likely to exhibit a turbulent regime involving IGWs at larger scales that would normally occur at smaller scales in other regions. This finding is consistent with the transition scale between balanced motions and waves found to be larger than 150-200 km by Qiu et al. (2018) in the NCAL region.

### 5.3 Seasonal dependence

Thirdly, we highlighted seasonal variations of the surface oceanic fine scales around New Caledonia. In the submesoscale-to-mesoscale range, temperature variance tends to increase during the cool season (June to November) compared to the warm season (December to May) whereas salinity variance decreases. However, only a weak seasonality is observed on the density variance suggesting that constructive effects between temperature and salinity also seasonally vary, with the relative importance of temperature diminishing during the warm season. We did not find any links with the seasonal variations of the upper 535 temperature and salinity profiles. Even if the density variance undergoes only weak seasonal variations, we found substantial variations of the APE at the surface with a larger reservoir during the cool season, mainly due to a less stratified water column during this season. Unfortunately, our dataset does not allow a separation of VSF into different seasons and we are unable to determine if the seasonality found on APE implies a seasonality on EKE and VSFs.



However, the seasonality of IGWs and SM motions has been shown to be out of phase near the surface, with SM motions
being the most energetic in late winter/early spring while IGWs are amplified during summer (Callies et al., 2015; Rocha
et al., 2016; Qiu et al., 2018; Lahaye et al., 2019). Our estimate of the surface APE is maximum during the cool season,
consistent with SM motions energised in late winter by mixed layer instabilities. Lahaye et al. (2019) demonstrate that the
amplification/dampening of IGWs at the surface compared to the interior is captured by a linear IGW model. Based on this
model, they estimate global maps of the horizontal KE ratio between the surface and the interior: the ratio of mode 1 shows
weak seasonality around New Caledonia, but higher modes (2, 3 and 4) have a higher ratio (i.e., amplification) during February.
We would then expect that surface VSFs would exhibit seasonal variations due to a change in SM KE fed by a larger reservoir
of APE during the cool season, as well as an amplification of IGW at the surface during the warm season. In the interior,
however, Qiu et al. (2018) showed that balanced motions are not likely to undergo substantial seasonal cycles around New
Caledonia whereas unbalanced motions will be more important in the winter/spring period. We would then expect that interior
VSFs around New Caledonia would exhibit seasonal variations in amplitude but would still exhibit similar slopes and divergent
contributions, characteristics of dynamics driven by IGWs.

## 5.4 Scale dependence

Finally, we showed different scale dependence in the submesoscale-to-mesoscale range. Using the scale-dependent Rossby
number, we found that 10 km is a cutoff scale under which the loss of balance is likely to occur. The Helmholtz decomposition
performed on VSFs globally show that ageostrophic motions are non negligible at scales smaller than 10 km as motions start to
have an important divergent component. The scale-dependent density ratio also showed that the dominant effect of temperature
on density slightly tends to diminish at smaller scales. ‡

The SF analysis performed here only characterised a continuum of scales without capturing the particular scales at which
the coherent internal tides vibrate. Those ones have well defined propagation and wavenumber, so that they can be captured
by along-track altimetry over a long record. Around New Caledonia, the first baroclinic mode associated with the M2 internal
tide is the most energetic (Vic et al., 2019; de Lavergne et al., 2019) and has wavelength between 120-150 km (Ray and Zaron,
2015; Zhao, 2018). Our SFs should not be impacted by the dominant wave (i.e., M2 mode 1) around New Caledonia. Because
SFs are averaged isotropically over bins of distance in our study, the signature of coherent internal tides is also likely to be
weak. However, the exact impact of coherent internal tides on structure functions have not been studied in the literature and
should be quantified by performing sensitivity analyses with idealised models including turbulence and waves. Such sensitivity
studies are beyond the scope of the current work and are left for future work.

## 5.5 Perspectives

Overall, our results suggest that the structure function analysis is a powerful tool that provides physically consistent results.
Amongst the benefits of the SF analysis, the capacity to be directly computed on uneven samples is clearly an advantage
compared to Fourier techniques. In order to provide more insights into the turbulent regimes at work, we tried to investigate the
direction of the KE cascade using third-order structure functions as done in Balwada et al. (2016) but our dataset revealed to be





too insufficient to compute accurate higher order statistics. We believe that more ADCP observations would solve this problem and provide better statistics in order to infer the direction of the energy cascades from SFs. A larger SADCP database would also allow disentangling seasonal and local variations in order to provide information on the seasonality of SFs and turbulent regimes.

Because of the substantial imprint of IGWs around New Caledonia, small-scale sea level features, that will be captured by the next generation of altimeters, will require additional work to disentangle wave signatures and geostrophic velocities. With the forthcoming SWOT satellite, an unprecedented opportunity exists to combine small-scale sea level measurements from space and in situ observing systems to better understand the small-scale ocean dynamics in regions of substantial KE and IGW activity, such as the New Caledonia region. We also believe in the support of high-resolution modelling to select a judicious deployment of in situ observations around New Caledonia and to provide valuable information about the link between sea level dynamics and the water column at fine scales.

*Code availability.* The code used to compute structure functions and analyse the results is made available on GitHub: https://github.com/serazing/serazin2019_scale-dependent

*Author contributions.* Guillaume Sérazin, Sophie Cravatte, Lionel Gourdeau and Frédéric Marin were funded by Institut of Recherche and Développement (IRD). Rosemary Morrow was funded by the University of Paul Sabatier (UPS), and Mei-Ling Dabat was funded by the Centre National de la Recherche Scientifique (CNRS) through the French TOSCA project. Guillaume Sérazin benefited of a IRD research grant for working on the SWOT project as well as financial support from the TOSCA/ROSES project. All authors contributed to a significant part to the presented scientific work.

*Competing interests.* The authors declare that they have no conflict of interest

*Acknowledgements.* The Ssalto/Duacs altimeter products were produced and distributed by the Copernicus Marine and Environment Monitoring Service (CMEMS) (http://www.marine.copernicus.eu). The authors acknowledge Collect Localisation Satellite for the M2 internal tide maps estimated from altimetry by Ray and Zaron (2015). The authors are grateful to all who contributed to the acquisition, processing and distribution of SADCP data: to the PIs of the cruises, the engineers, and the crew. They wish to thank in particular the R/V Alis crew, and E. Firing, J. Hummon and P. Caldwell for maintaining the Joint Archive for Shipboard ADCP (JASADCP, http://ilikai.soest.hawaii.edu/sadcp/main-inv.html). The ADCP dataset produced after quality control at the Laboraboire d'Études en Géophysique et Océanographie Spatiale is available on request. Sea surface salinity data derived from voluntary observing ships were collected, validated, archived, and made freely available by the French Sea Surface Salinity Observation Service (http://www.legos.obs-mip.fr/observations/sss/)"



Argo data were collected and made freely available by the International Argo Program and the national programs that contribute to it.

600    (http://www.argo.ucsd.edu, http://argo.jcommops.org). The Argo Program is part of the Global Ocean Observing System.

The authors wish to acknowledge D. Balwada for fruitful discussions on the use of structure functions with ocean observations and Les Houches School of Physics for hosting summer schools allowing such interactions between scientists.



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



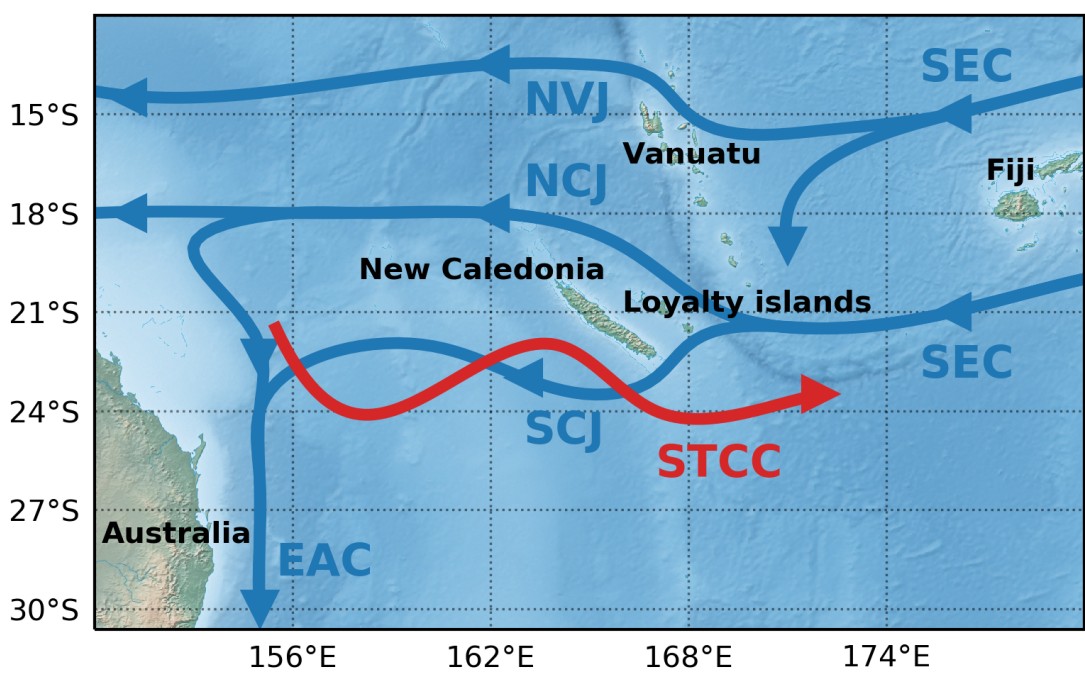

**Figure 1.** Geography and circulation around New Caledonia reproduced from Cravatte et al. (2015), with the name of islands (black names), thermocline currents (blue curves and names) and surface currents (red curves and names). Indicated are the SEC (South Equatorial Current), NVJ (North Vanuatu Jet), NCJ and SCJ (North and South Caledonian Jet), STCC (SubTropical CounterCurrent), EAC (East Australian Current).



**Figure 2.** Regions of study (red boxes) around New Caledonia: Eddy Kinetic Energy estimated using DUACS (top left), M2 coherent internal tide estimated from satellite altimetry (top right), SADCP (bottom left) and TSG (bottom right) sections performed in the South West Pacific.



**Figure 3.** Mean seasonal profiles of temperature (blue), salinity (red), potential density $\sigma_0$ referenced to the surface pressure (green) and buoyancy frequency $N^2$ (orange), separated into a cool season (June to November, solid curves) and a warm season (December to May, dashed curves).







**Figure 4.** Mean velocity structure functions computed on each segment in the VAUB (left) and ECAL (right) regions plotted in the range 3-100 km. Longitudinal and transverse structure functions (not shown) are used to compute the total SFs (green curve) as well as the associated rotational (red) and divergent (purple) SFs. The total structure functions are also shown for each segment (shaded gray curves) inside the region of study. Classic power laws ($r^2$, $r$, $r^{2/3}$) are plotted for reference with oceanic turbulence theories as well as the Garrett and Munk spectrum structure function (GM81). Colour shading corresponds to confidence intervals at 5% and 95%, estimated from a bootstrap method.





**Figure 5.** Same as Fig. 4 but for the NCAL (left) and SCAL (right) regions.

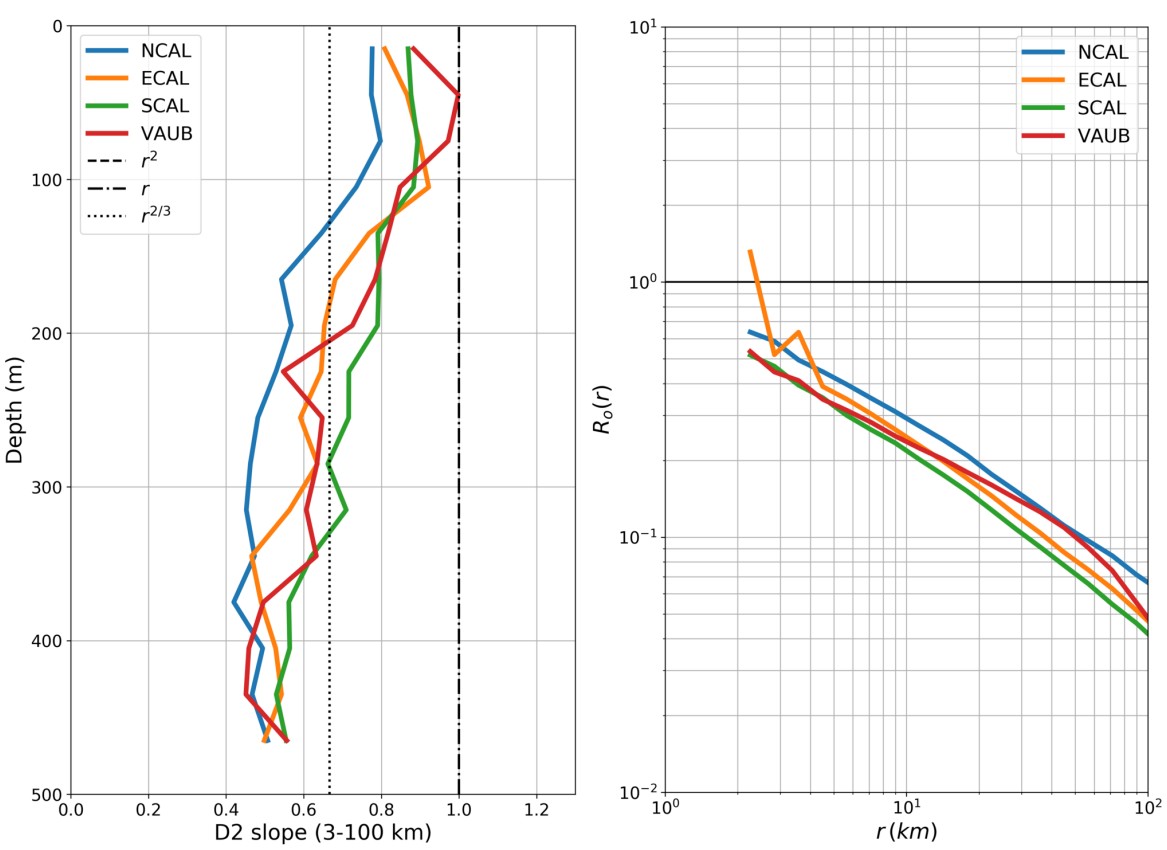

**Figure 6.** Evolution of the total structure function slope with depth for the range 3-100 km (left panel). Classic power laws ($r^2$, $r$, $r^{2/3}$) are plotted for reference with oceanic turbulence theories. Scale-dependent Rossby number $R_o$ estimated from the total structure function averaged over depth (right panel).



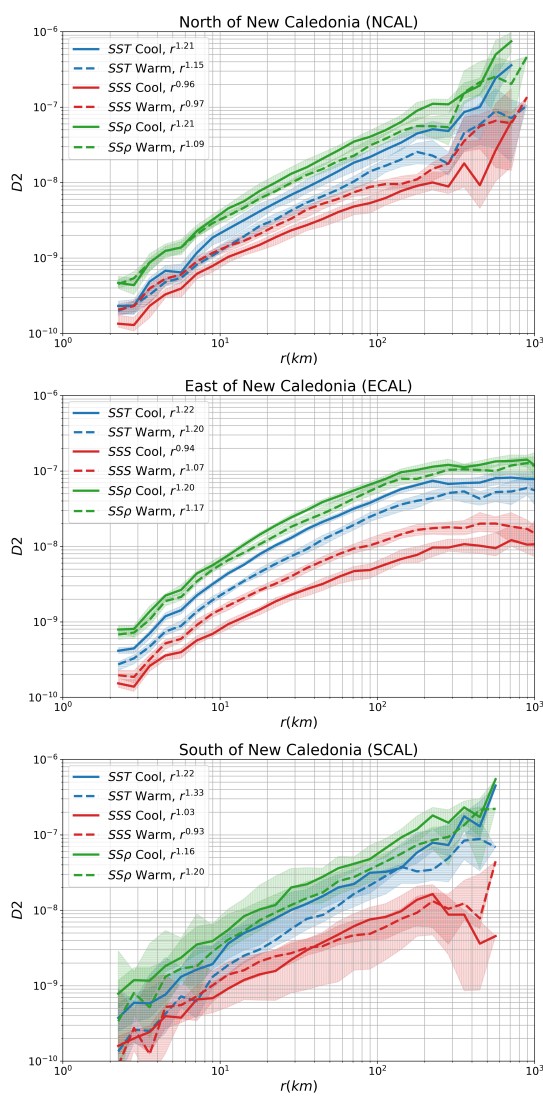

**Figure 7.** Second order structure functions computed for Sea Surface Salinity ($SSS$, red curves), Sea Surface Temperature ($SST$, blue curves), Sea Surface Density ($SS\rho$, green curves) in three regions surroundings New Caledonia: NCAL (top), ECAL(middle), SCAL(bottom). Seasonality is shown between cool (June to November, solid curves) and warm (December to May, dashed curves) seasons. Values of the structure function slopes, estimated between 3 and 100 km, are given in the caption. Colour shading corresponds to confidence intervals at 5% and 95%, estimated from a bootstrap method.


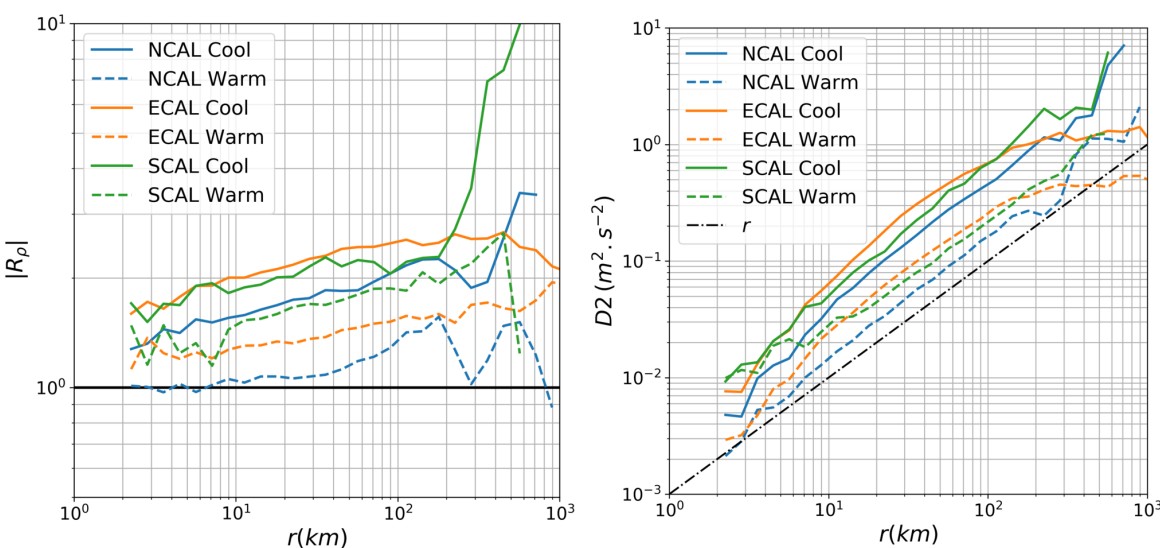

**Figure 8.** Absolute density ratio $|R_\rho|$ (right panel) and structure function of Available Potential Energy (left panel) as a function of the separation vector $r$ for the regions NCAL (blue), ECAL (orange) and SCAL (green).Seasonality is shown between cool (June to November, solid curves) and warm (December to May, dashed curves) seasons. In the left panel, the black horizontal line corresponds to $|R_\rho| = 1$: above this line temperature dominates density variation. In the right panel, the dashed dotted black line correspond to the power law $r^1$.