# Peer review of "Scale-dependent analysis of in situ observations in the mesoscale to submesoscale range around New Caledonia"

_Ocean Science, 2019_

## Referee Comment (RC1) · Anonymous Referee #1 · 24 Feb 2020

The authors conduct a fairly extensive analysis of a complex region near New Caledonia in regards to scale-dependent characterization of turbulence characteristics while also considering seasonal and depth dependence. The most impressive part of the study is that it is done using a mixture of observational data (ship-based ADCPs, ship-based temperature and salinity), as opposed to modeling. The analysis is conducted by considering possible approaches to extract the most from upcoming SWATH missions. Clearly the authors spend significant effort on this. The paper is very well written and makes good use of past literature. I am aware of two papers in which similar structure function approaches have been used, with some success in extracting the direction of energy cascade from drifter data:

[Figure]

Poje, A.C., et al., 2017: Evidence of a forward energy cascade and Kolmogoroff self-similariy in submesoscale ocean surface drifter observations. Physics of Fluids, 29, 020701.

Mensa, J.A., et al., 2018: Surface drifter observations from the Arctic Ocean's Beaufort Sea: evidence of submesoscale dynamics. JGR-Oceans, 123/4, 2635-2645.

which are not cited here. I encourage the authors to take a look.

The resulting picture painted by the analysis is complex, as expected in a region consisting by mesoscale motions, deep waves and wakes behind islands. In many ways, this is the real strength of the paper.

I just have one major question: in some of observational (not modeling) studies in which submesoscales have been truly detected, for instance (which are not cited by the way):

D'Asaro et al., 2018: Ocean convergence and dispersion of flotsam. PNAS, https://doi.org/10.1073/pnas.1718453115.

Poje, A.C., et al 2014: Submesoscale dispersion in the vicinity of the Deepwater Horizon spill. Proc. Nat. Acad. Sci. 111, 12693-12698.

submesoscale flows exist because they are not overrun by mesoscale flows, which are far stronger. So, how come the authors emphasize weak submesoscale motions in a region presumably dominated by mesoscale and other very strong flows (IGWs and wakes)? Or submesoscale band is simply a generic name for many other phenomena that map into that range somehow?

---

## Referee Comment (RC2) · Haijin Cao (Referee) · 24 Feb 2020

**Review comments**

This study analyzed the data from ship-based ADCP and TSG data using second order structural functions to understand the dynamics of the currents in the Southwest Pacific Ocean. A Helmholtz decomposition was used to decompose the rotational and divergent components. One of the main points was to explore the possible regimes for the submesoscale range. It is a very interesting study and I am suggesting minor revisions for this study, and my suggestions to change are listed below.

Itemed suggestions

1. The instruction is a little bit lengthy, but missed some very relative publications (e.g., Torres et al., 2018 (JGR-Oceans); Cao et al., 2019 (JGR-Oceans); Pearson et al., 2019 (JGR-Oceans)). The key point of this study is to examine the role of rotational and divergent motions using the spectral analysis method. It is not necessary to talk too many details about the study for other sea regions, since they used the similar research method.

2. Around Line 25. It is not appropriate to say "IG waves include internal tides". Though the IG waves have a frequency range that includes the tidal frequencies, they have different generating regimes. Besides, IG waves have their own dispersion relationship rather than at fixed tidal frequencies. Please modify this sentence.

3. Figure 4 shows that the confidence intervals are extremely large for the decomposed components. Are the results still reliable? We ever tried the decomposition in the structural functions and found out that sometimes one of the components could fall into negative values because the other components dominated. That could be the problem. It is necessary to discuss the problem of the method and the results could partly be trusted (maybe the dominated component).

4. It is interesting that no substantial seasonal variation is noticed. Usually, the seasonality of submesoscale motions should be prominent. Please make some comments on this.

5. In the text, sometimes you use figure 7 and sometimes Fig. 7.

---

## Author Comment (AC1) · 22 Mar 2020

Thank you very much for you comments. Please find our response to your different points here.

1. We do think that providing the results found in other sea regions that used similar techniques are important in the introduction. The main reason is because we find different conclusions compared to those studies, thus highlighting the regional variability of submesoscale and internal wave processes in particular in the region around New Caledonia.

[Figure]

2. We agree that internal waves vibrate at tidal frequencies, but they do follow the same physics and the same dispersion relationship as internal waves. As the frequency of internal tides is fixed, so is the wavenumber for a given vertical mode. Please compare your statement with GFD lessons such as those found at this link https://gfd.whoi.edu/wp-content/uploads/sites/18/2018/03/lecture06$_2$1356.$pdf$

3. In the two upper layers presented in Figure 4, only the lower confidence intervals are quite large for the divergent component but it means that in certain case the divergent component may be very weak, though the rotational components still dominate. We think it is better to focus on the upper confidence interval of the divergent component and the lower confidence interval of the rotational component. If they do not overlap, we conclude for a clean separation between the divergent and the rotational component. It becomes different in the interior layer where both confidence interval for divergent and rotational motions overlap, meaning that the two types of motions are important. For those reasons, we think the results are reliable and can still be interpreted as we do in the paper.

4. We should emphasise that we do not see any seasonal variation in the density variance but there are seasonal variations in temperature and salinity variance. Because we do not have enough data to look at the seasonality of velocity structure functions, it is hard to discuss the lack of seasonality of submesoscale motions in general. Perhaps there is none because we are here in the tropics and the seasonal variations are less marked than at midlatitudes. Perhaps there is some but we do not have any support to discuss it.

5. Thanks, we have homogenised the document to use only Fig.

---

## Author Comment (AC2) · 22 Mar 2020

Thanks for your review and the two additional papers on the structure functions you suggested. The two study you mentioned mainly use drifter data as in Balwada et al. (2016), and are therefore able to estimate third order structure functions. We found that we were lacking a sufficient number of observations from ADCPs to estimate higher order statistics.

Here is a response to your question: "Submesoscale flows exist because they are not overrun by mesoscale flows, which are far stronger. So, how come the authors emphasize weak submesoscale motions in a region presumably dominated by mesoscale and

other very strong flows (IGWs andwakes)? Or submesoscale band is simply a generic name for many other phenomena that map into that range somehow?"

The definition of submesoscale flows in terms of fluid dynamics is: flows with a Rossby number $Ro$ close to the Froude number $Fr$ (i.e. a burger number $Bu$ close to 1). In other words, both stratification and rotation are important in the dynamics of submesoscale motions. In the ocean those conditions are generally met for horizontal spatial scales of 100 m to 10 km, and vertical scales of 10 m to 10 km. Yet, this depends on the regions because the Coriolis force is latitudinally dependent and the stratification is regionally dependent, but the scientific community usually refer to scales smaller than 10 km as being in the subsmesoscale band. Because of the flow is turbulent, there is a continuum between mesoscale and submesoscale motions and energy is transferred between scales in a continuous manner. The result is a continuous spectrum with energy dropping as scales get smaller. Thus, I will not say that submesoscale are overrun by mesoscale flows because they are a continuation of mesoscale eddies and the energy pathway to smaller scales.

Mesoscale eddies are a potential source of submesoscale structures as they are associated with strong density fronts storing available potential energy, from which instabilities (mixed layer instability, frontogenesis and turbulent thermal wind) feed submesoscale motions. When mesoscale eddies are weak, it is still possible to have submesoscale vortices generated by sharp but well elongated density fronts through the same types of instabilities as shown in D'Asaro et al., 2018. In this case, it is easy to isolate submesoscale structures as they are not dependent of transient features like mesoscale eddies.

---

## Author Response (AR2)

**Response to reviewers (second revision)**

June 3, 2020

**1   General remarks**

We thank the two reviewers for their recommendations to publish this manuscript and for their comments. Following the reviewers' suggestions, we have added a short discussion in the manuscript about the validity of the hypothesis we used for the computation of structure functions.

**2   Detailed answers to Reviewer 2**

> The large error likely results from the violations of the assumptions, which makes it implausible. The authors have to assess to what degree the assumptions are violated.

We have discussed the validity of these assumptions at the end of the manuscript as well as how we could check these assumptions using a high-resolution modelling approach.

**Response to reviewers**

March 22, 2020

**1 General remarks**

We thank the two reviewers for their recommendations to publish this manuscript and for their comments. Following the reviewer suggestions, we only homogenised the term Figure by Fig in the latest version of the paper, so we do not find necessary to join the marked-up manuscript here.

**2 Detailed answers to Reviewer 1**

**2.1 Minor comments**

> I am aware of two papers in which similar structure function approaches have been used, with some success in extracting the direction of energy cascade from drifter data:
>
> - Poje, A.C., et al., 2017: Evidence of a forward energy cascade and Kolmogoroff self-similariy in submesoscale ocean surface drifter observations. Physics of Fluids, 29,020701.
>
> - Mensa, J.A., et al., 2018: Surface drifter observations from the Arctic Ocean?s BeaufortSea: evidence of submesoscale dynamics. JGR-Oceans, 123/4, 2635-2645.
>
> which are not cited here. I encourage the authors to take a look.

Thanks for your review and the two additional papers on the structure functions you suggested. The two studies you mentioned mainly use drifter data as in Balwada et al.(2016), and are therefore able to estimate third order structure functions. We found that we were lacking a sufficient number of observations from ADCPs to estimate higher order statistics.

> I just have one major question: in some of observational (not modeling) studies in which submesoscales have been truly detected, for instance (which are not cited by the way):
>
> - D'Asaro et al., 2018: Ocean convergence and dispersion of flotsam.PNAS,https://doi.org/10.1073/pnas.1718453115.
>
> - Poje, A.C., et al 2014: Submesoscale dispersion in the vicinity of the Deepwater Horizon spill. Proc. Nat. Acad. Sci. 111, 12693-12698.
>
> Submesoscale flows exist because they are not overrun by mesoscale flows, which are far stronger. So, how come the authors emphasize weak submesoscale motions in a region presumably dominated by mesoscale and other very strong flows (IGWs and wakes)? Or submesoscale band is simply a generic name for many other phenomena that map into that range somehow?

Thanks for your review and the two additional papers on the structure functions you suggested. The two study you mentioned mainly use drifter data as in Balwada et al.(2016), and are therefore

able to estimate third order structure functions. We found that we were lacking a sufficient number of observations from ADCPs to estimate higher order statistics.

The definition of submesoscale flows in terms of fluid dynamics is: flows with a Rossby number $Ro$ close to the Froude number $Fr$ (i.e. a burger number $Bu$ close to 1). In other words, both stratification and rotation are important in the dynamics of submesoscale motions. In the ocean those conditions are generally met for horizontal spatial scales of 100 m to 10 km, and vertical scales of 10 m to 10 km. Yet, this depends on the regions because the Coriolis force is latitudinally dependent and the stratification is regionally dependent, but the scientific community usually refer to scales smaller than10 km as being in the submesoscale band. Because of the flow is turbulent, there is a continuum between mesoscale and submesoscale motions and energy is transferred between scales in a continuous manner. The result is a continuous spectrum with energy dropping as scales get smaller. Thus, I will not say that submesoscale are overrun by mesoscale flows because they are a continuation of mesoscale eddies and the energy pathway to smaller scales.Mesoscale eddies are a potential source of submesoscale structures as they are associated with strong density fronts storing available potential energy, from which in-stabilities (mixed layer instability, frontogenesis and turbulent thermal wind) feed sub-mesoscale motions. When mesoscale eddies are weak, it is still possible to have submesoscale vortices generated by sharp but well elongated density fronts through the same types of instabilities as shown in D'Asaro et al., 2018. In this case, it is easy to isolate submesoscale structures as they are not dependent of transient features like

**3   Detailed answers to Reviewer 2**

**3.1   Minor comments**

> 1. The instruction is a little bit lengthy but misse d some very relative publications ( e.g., Torres et al., 2018 JGR Oceans )); Cao et al., 2019 (JGR Oceans )); Pearson et al., 2019 ( JGR Oceans The key point of this study is to examine the role of rotational and divergent motions us ing the spectral analysis method. It is not necessary to talk too many details about the study for other sea regions, since they used the similar research method.

We do think that providing the results found in other sea regions that used similar techniques are important in the introduction. The main reason is because we find different conclusions compared to those studies, thus highlighting the regional variability of submesoscale and internal wave processes in particular in the region around NewCaledonia.

> 2. Around Line 25. It is not appropriate to say "IG waves include internal tides". Though the IG waves have a frequency range that includes the tidal frequencies, they have different generating regimes. Besides, IG waves have their own dispersion relationship rather than at fixed tidal frequencies. Please modify this sentence.

We agree that internal waves vibrate at tidal frequencies, but they do follow the same physics and the same dispersion relationship as internal waves. As the fre-quency of internal tides is fixed, so is the wavenumber for a given vertical mode.Please compare your statement with GFD lessons such as those found at this link https://gfd.whoi.edu/wp-content/uploads/sites/18/2018/03/lecture0621356.pdf

> 3. Figure 4 shows that the confidence intervals are extremely large for the decomposed components. Are the results still reliable? We ever tried the decomposition in the structural functions and found out that sometimes one of the components could fall into negative values because the other components dominated. That could be the problem. It is necessary to discuss the problem of the method and the results could partly be trusted maybe the dominated component).

In the two upper layers presented in Figure 4, only the lower confidence intervals are quite large for the divergent component but it means that in certain case the divergent component may be very weak, though the rotational components still dominate. We think it is better to focus on the upper confidence interval of the divergent component and the lower confidence interval of the rotational component. If they do not overlap, we conclude for a clean separation between the divergent and the rotational component. It becomes different in the interior layer where both confidence interval for divergent and rotational motions overlap, meaning that the two types of motions are important. For those reasons, we think the results are reliable and can still be interpreted as we do in the paper.

4. It is interesting that no substantial seasonal variation is noticed. Usually, the seasonality of submesoscale motions should be prominent. Please make some comments on this.

We should emphasise that we do not see any seasonal variation in the density variance but there are seasonal variations in temperature and salinity variance. Because we do not have enough data to look at the seasonality of velocity structure functions, it is hard to discuss the lack of seasonality of submesoscale motions in general. Perhaps there is none because we are here in the tropics and the seasonal variations are less marked than at midlatitudes. Perhaps there is some but we do not have any support to discuss it.

5. In the text, sometimes you use figure 7 and sometimes Fig. 7.

Thanks, we have homogenised the document to use only Fig.

[revised manuscript text omitted]